# Hierarchical Subtask Discovery with Non-Negative Matrix Factorization

**Adam C. Earle**
Department of Computer Science and Applied Mathematics
University of the Witwatersrand
Johannesburg, South Africa
`adam.earle@students.wits.ac.za`

**Andrew M. Saxe**
Center for Brain Science
Harvard University
MA, USA
`asaxe@fas.harvard.edu`

**Benjamin Rosman**
Council for Scientific and Industrial Research
Pretoria, South Africa, and
Department of Computer Science and Applied Mathematics
University of the Witwatersrand
Johannesburg, South Africa
`brosman@csir.co.za`

## Abstract

Hierarchical reinforcement learning methods offer a powerful means of planning flexible behavior in complicated domains. However, learning an appropriate hierarchical decomposition of a domain into subtasks remains a substantial challenge. We present a novel algorithm for subtask discovery, based on the recently introduced multitask linearly-solvable Markov decision process (MLMDP) framework. The MLMDP can perform never-before-seen tasks by representing them as a linear combination of a previously learned basis set of tasks. In this setting, the subtask discovery problem can naturally be posed as finding an optimal low-rank approximation of the set of tasks the agent will face in a domain. We use non-negative matrix factorization to discover this minimal basis set of tasks, and show that the technique learns intuitive decompositions in a variety of domains. Our method has several qualitatively desirable features: it is not limited to learning subtasks with single goal states, instead learning distributed patterns of preferred states; it learns qualitatively different hierarchical decompositions in the same domain depending on the ensemble of tasks the agent will face; and it may be straightforwardly iterated to obtain deeper hierarchical decompositions.

## 1 Introduction

Hierarchical reinforcement learning methods hold the promise of faster learning in complex state spaces and better transfer across tasks, by exploiting planning at multiple levels of detail (Barto & Madadevan, 2003). A taxi driver, for instance, ultimately must execute a policy in the space of torques and forces applied to the steering wheel and pedals, but planning directly at this low level is beset by the curse of dimensionality. Algorithms like HAMS, MAXQ, and the options framework permit powerful forms of hierarchical abstraction, such that the taxi driver can plan at a higher level, perhaps choosing which passengers to pick up or a sequence of locations to navigate to (Sutton et al., 1999; Dietterich, 2000; Parr & Russell, 1998). While these algorithms can overcome the curse of dimensionality, they require the designer to specify the set of higher level actions or subtasks available

to the agent. Choosing the right subtask structure can speed up learning and improve transfer across tasks, but choosing the wrong structure can slow learning (Solway et al., 2014; Brunskill & Li, 2014). The choice of hierarchical subtasks is thus critical, and a variety of work has sought algorithms that can automatically discover appropriate subtasks.

One line of work has derived subtasks from properties of the agent's state space, attempting to identify states that the agent passes through frequently (Stolle & Precup, 2002). Subtasks are then created to reach these bottleneck states (van Dijk & Polani, 2011; Solway et al., 2014; Diuk et al., 2013). In a domain of rooms, this style of analysis would typically identify doorways as the critical access points that individual skills should aim to reach (Şimşek & Barto, 2009). This technique can rely only on passive exploration of the agent, yielding subtasks that do not depend on the set of tasks to be performed, or it can be applied to an agent as it learns about a particular ensemble of tasks, thereby suiting the learned options to a particular task set.

Another line of work converts the target MDP into a state transition graph. Graph clustering techniques can then identify connected regions, and subtasks can be placed at the borders between connected regions (Mannor et al., 2004). In a rooms domain, these connected regions might correspond to rooms, with their borders again picking out doorways. Alternately, subtask states can be identified by their betweenness, counting the number of shortest paths that pass through each specific node (Şimşek & Barto, 2009; Solway et al., 2014). Other recent work utilizes the eigenvectors of the graph laplacian to specify dense rewards for option policies that are defined over the full state space (Machado et al., 2017). Finally, other methods have grounded subtask discovery in the information each state reveals about the eventual goal (van Dijk & Polani, 2011). Most of these approaches aim to learn options with a single or low number of termination states, can require high computational expense (Solway et al., 2014), and have not been widely used to generate multiple levels of hierarchy (but see Vigorito & Barto (2010); McNamee et al. (2016)).

Here we describe a novel subtask discovery algorithm based on the recently introduced Multitask linearly-solvable Markov decision process (MLMDP) framework (Saxe et al., 2017), which learns a basis set of tasks that may be linearly combined to solve tasks that lie in the span of the basis (Todorov, 2009a). We show that an appropriate basis can naturally be found through non-negative matrix factorization (Lee & Seung, 1999; 2000), yielding intuitive decompositions in a variety of domains. Moreover, we show how the technique may be iterated to learn deeper hierarchies of subtasks.

In line with a number of prior methods, (Solway et al., 2014; McNamee et al., 2016) our method operates in the batch off-line setting; with immediate application to probabilistic planning. The subtask discovery method introduced in Machado et al. (2017), which also utilizes matrix factorization techniques to discover subtasks albeit from a very different theoretical foundation, is notable for its ability to operate in the online RL setting, although it is not immediately clear how the approach taken therein might achieve a deeper hierarchical architecture, or enable immediate generalization to novel tasks.

## 2 BACKGROUND: THE MULTITASK LMDP

In the multitask framework of Saxe et al. (2017), the agent faces a set of tasks where each task has an identical transition structure, but different terminal rewards, modeling the setting where an agent pursues different goals in the same fixed environment. Each task is modeled as a finite-exit LMDP (Todorov, 2009a). The LMDP is an alternative formulation of the standard MDP that carefully structures the problem formulation such that the Bellman optimality equation becomes linear in the exponentiated cost-to-go. As a result of this linearity, optimal policies compose naturally: solutions for rewards corresponding to linear combinations of two optimal policies are simply the linear combination of their respective exponentiated cost-to-go functions (Todorov, 2009b). This special property of LMDPs is exploited by Saxe et al. (2017) to develop a multitask reinforcement learning method that uses a library of basis tasks, defined by their boundary rewards, to perform a potentially infinite variety of other tasks–any tasks that lie in the subspace spanned by the basis can be performed optimally.

Briefly, the LMDP (Todorov, 2009a;b) is defined by a three-tuple $L = \langle S, P, R \rangle$, where $S$ is a set of states, $P$ is a passive transition probability distribution $P : S \times S \to [0, 1]$, and $R$ is an

expected instantaneous reward function $R : S \to \mathbb{R}$. The 'action' chosen by the agent is a full transition probability distribution over next states, $a(\cdot|s)$. A control cost is associated with this choice such that a preference for energy-efficient actions is inherently specified: actions corresponding to distributions over next states that are very different from the passive transition probability distribution are expensive, while those that are similar are cheap. In this way the problem is regularized by the passive transition structure. Finally, the LMDP has rewards $r_i(s)$ for each interior state, and $r_b(s)$ for each boundary state in the finite exit formulation. The LMDP can be solved by finding the *desirability* function $z(s) = e^{V(s)/\lambda}$ which is the exponentiated cost-to-go function for a specific state $s$. Here $\lambda$ is a temperature-like parameter related to the stochasticity of the solution. Given $z(s)$, the optimal control can be computed in closed form (see Todorov (2007) for details). Despite the restrictions inherent in the formulation, the LMDP is generally applicable; see the supplementary material in Saxe et al. (2017) for examples of how the LMDP can be applied to non-navigational, and conceptual tasks.

A primary difficulty in translating standard MDPs into LMDPs is the construction of the action-free passive dynamics $P$ (although a general way of approximating MDPs using LMDPs is given in Todorov (2007)); however, in many cases, this can simply be taken as the resulting Markov chain under a uniformly random policy. In this instance the problem is said to be 'entropy regularized'. A similar problem set-up appears in a number of recent works (Schulman et al., 2017; Haarnoja et al., 2017).

The Multitask LMDP (MLDMP) (Saxe et al., 2017) operates by learning a set of $N_t$ tasks, defined by LMDPs $L_t = \langle S, P, q_i, q_b^t \rangle$, $t = 1, \cdots, N_t$ with identical state space, passive dynamics, and internal rewards, but different instantaneous exponentiated boundary reward structures $q_b^t = \exp(r_b^t/\lambda)$, $t = 1, \cdots, N_t$. The set of LMDPs represent an ensemble of tasks with different ultimate goals. We can define the task basis matrix $Q = \begin{bmatrix} q_b^1 & q_b^2 & \cdots & q_b^{N_t} \end{bmatrix}$ consisting of the different exponentiated boundary rewards. Solving these LMDPs gives a set of desirability functions $z_i^t$, $t = 1, \cdots, N_t$ for each task, which can be formed into a desirability basis matrix $Z = \begin{bmatrix} z_i^1 & z_i^2 & \cdots & z_i^{N_t} \end{bmatrix}$ for the multitask module. With this machinery in place, if a new task with boundary reward $q$ can be expressed as a linear combination of previously learned tasks, $q = Qw$. Then the same weighting can be applied to derive the corresponding optimal desirability function, $z = Zw$, due to the compositionality of the LMDP. More generally, if the new task cannot be exactly expressed as a linear combination of previously learned tasks, a significant jump-start in learning may nevertheless be gained by finding an approximate representation.

## 2.1 STACKING THE MLMDP

The multitask module can be stacked to form deep hierarchies (Saxe et al., 2017) by iteratively constructing higher order MLMDPs in which higher levels select the instantaneous reward structure that defines the current task for lower levels in a feudal-like architecture. This recursive procedure is carried out by firstly augmenting the layer $l$ state space $\tilde{S}^l = S^l \cup S_t^l$ with a set of $N_t$ terminal boundary states $S_t^l$ called *subtask* states. Transitioning into a subtask state corresponds to a decision by the layer $l$ MLMDP to access the next level of the hierarchy, and is equivalent to entering a state of the higher layer. These subtask transitions are governed by a new $N_t^l$-by-$N_i^l$ passive dynamics matrix $P_t^l$. In the augmented MLMDP, the full passive dynamics are taken to be $\tilde{P}^l = [P_i^l; P_b^l; P_t^l]$, corresponding to transitions to interior states, boundary states, and subtask states respectively. Transitions dynamics for the higher layer $[P_i^{l+1}; P_b^{l+1}]$ are then suitably defined (Saxe et al., 2017). Solving the higher layer MLMDP will yield an optimal action $a(\cdot|s)$ making some transitions more likely than they would be under the passive dynamic, indicating that they are more desirable for the current task. Similarly, some transitions will be less likely than they would be under the passive dynamic, indicating that they should be avoided for the current task. The instantaneous rewards for the lower layer are therefore set to be proportional to the difference between the controlled and passive dynamic, $r_t^l \propto a_i^{l+1}(\cdot|s) - p_i^{l+1}(\cdot|s)$. See Fig.(1) for more details.

Crucially, in order to stack these modules, both the subtask states themselves $S_t^l$, and the new passive dynamic matrix $P_t^l$ must be defined. These are typically hand crafted at each level. A key contribution of this paper is to make these processes autonomous.

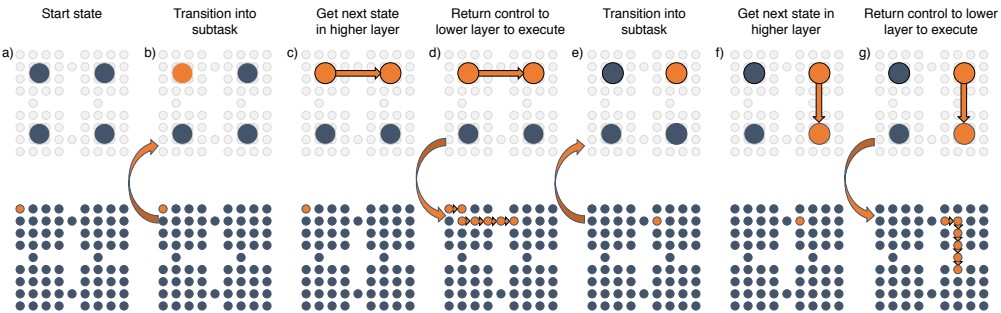

Figure 1: Execution model for the hierarchical MLMDP (Saxe et al., 2017). a) Beginning at some start state, the agent will make a transition under $\tilde{P}^1$. This transition may be to an interior, boundary, or subtask state. b) Transitioning into a subtask state is equivalent to entering a state of the higher layer MLMDP. No 'real' time passes during this transition. c) The higher layer MLMDP is then solved and a next higher layer state is drawn. d) Knowing the next state at the higher layer allows us to specify the reward structure defining the current task at the lower layer. Control is then passed back to the lower layer to achieve this new task. Notice that the details of how this task should be solved are left to the lower layer (one possible trajectory being shown). e) At some point in the future the agent may again elect to transition into a subtask state - in this instance the transition is into a different subtask corresponding to a different state in the higher layer. f) The higher layer MLMDP is solved, and a next state drawn. This specifies the reward structure for a new task at the lower layer. g) Control is again passed back to the lower layer, which attempts to solve the new task. This process continues until the agent transitions into a boundary state.

## 3 SUBTASK DISCOVERY VIA NON-NEGATIVE MATRIX FACTORIZATION

Prior work has assumed that the task basis $Q$ is given *a priori* by the designer. Here we address the question of how a suitable basis may be learned. A natural starting point is to find a basis that retains as much information as possible about the ensemble of tasks to be performed, analogously to how principal component analysis yields a basis that maximally preserves information about an ensemble of vectors. In particular, to perform new tasks well, the desirability function for a new task must be representable as a (positive) linear combination of the desirability basis matrix $Z$. This naturally suggests decomposing $Z$ using PCA (i.e., the SVD) to obtain a low-rank approximation that retains as much variance as possible in $Z$. However, there is one important caveat: the desirability function is the exponentiated cost-to-go, such that $Z = \exp(V/\lambda)$. Therefore $Z$ must be non-negative, otherwise it does not correspond to a well-defined cost-to-go function.

Our approach to subtask discovery is thus to uncover a low-rank representation through non-negative matrix factorization, to realize this positivity constraint (Lee & Seung, 1999; 2000). We seek a decomposition of $Z$ into a data matrix $D \in \mathbf{R}^{(m \times k)}$ and a weight matrix $W \in \mathbf{R}^{(k \times n)}$ as:

$$Z \approx DW, \tag{1}$$

where $d_{ij}, w_{ij} \geq 0$. The value of $k$ in the decomposition must be chosen by a designer to yield the desired degree of abstraction, and is referred to as the *decomposition factor*. A small value of $k$ corresponds to a high degree of abstraction since the variance in the desirability space $Z$ must be captured in a $k$ dimensional subspace spanned by the vectors in the data matrix $D$. Conversely, a large value of $k$ corresponds to a low degree of abstraction.

Since $Z$ is strictly positive, the non-negative decomposition is not unique for any value of $k$ (Donoho & Stodden, 2004). Formally then, we seek a decomposition which minimizes the cost function

$$d_\beta(Z||DW), \tag{2}$$

where $d$ denotes the $\beta$-divergence, a subclass of the more familiar Bregman Divergences (Hennequin et al., 2011), between the true basis $Z$ and the approximate basis $D$. The $\beta$-divergence collapses

to the better known statistical distances for $\beta \in \{1, 2\}$, corresponding to the Kullback-Leibler and Euclidean distances respectively (Cichocki et al., 2011).

Crucially, since $Z$ depends on the set of tasks that the agent will perform in the environment, the representation is defined by the tasks taken against it, and is not simply a factorization of the domain structure. To keep the focus on the decomposition strategy, we assume, here and throughout, that $Z \in \mathbf{R}^{n \times n}$ is given. The basis set of tasks can be a tiny fraction of the set of possible tasks in the space. As an example, suppose we consider tasks with boundary rewards at any of two separate locations in an $n$-dimensional world such that there are $n$-choose-2 possible tasks (corresponding to tasks like 'navigate to point A or B'). We require only an $n$-dimensional $Z$ matrix containing tasks to navigate to each point individually. The resulting subtasks we uncover will aid in solving all of these $n$-choose-2 tasks. More generally we might consider tasks in which boundary rewards are placed at three or more locations, etc. To know $Z$ therefore means to know an optimal policy to achieve $n$ of $\sim 2^n$ tasks in a space. An online version of this method would estimate $Z$ from data, either directly or by learning a transition model (see Machado et al. (2017) for some possibilities).

## 3.1 CONCEPTUAL DEMONSTRATION

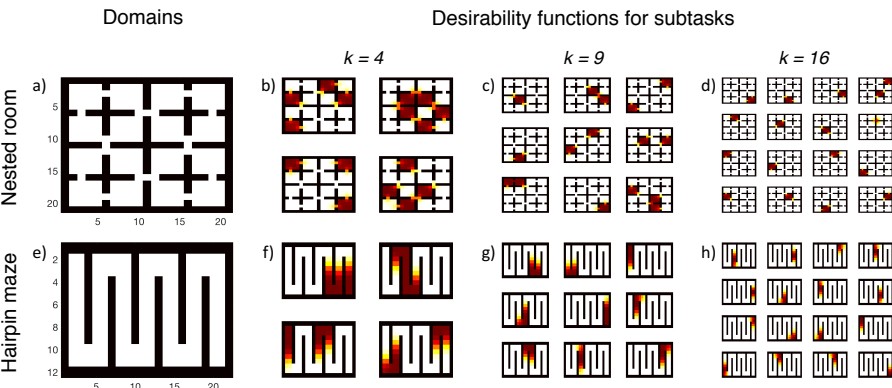

Figure 2: Intuitive decompositions in structured domains. All colour-plots correspond to the desirability functions for subtasks overlaid onto the base domains shown in panels a) and e). b,c,d) Subtasks correspond to 'regions', distributed patterns over preferred states, rather than single states. Where the decomposition factor is chosen to match the structure of the domain (here $k = 16$ for example), subtasks correspond to an intuitive semantic - "go to room X". f,g,h) Again, subtasks correspond to regions rather than single states. Collectively the subtasks form an approximate cover for the space.

To demonstrate that the proposed scheme recovers an intuitive decomposition, we consider the resulting low-rank approximation to the desirability basis in two domains, for a few hand-picked decomposition factors. All results presented in this section correspond to solutions to Eqn.(2) for $\beta = 1$ so that the cost function is taken to be the KL-divergence (although the method does not appear to be overly sensitive to $\beta \in [1, 2]$). Note that in the same way that the columns of $Z$ represent the exponentiated cost-to-go for the single-states tasks in the basis, so the columns in $D$ represent the exponentiated cost-to-go for the discovered subtasks.

In Fig. 2, we compute the data matrix $D \in \mathbf{R}^{m \times k}$ for $k = \{4, 9, 16\}$ for both the nested rooms domain, and the hairpin domain. The desirability functions for each of the subtasks is then plotted over the base domain. All of the decompositions share a number of properties intrinsic to the proposed scheme. Most notably, the subtasks themselves do not correspond to single states (like bottle-neck states), but rather to complex distributions over preferred states. By way of example, semantically, a single subtask in Fig. 2-d corresponds to the task 'Go to Room', where any state in the room is suitable as a terminal state for the subtask. Also, since $Z$ is taken to be the full basis matrix in this example, the distributed patterns of the subtasks collectively form an approximate cover for the full space. This is true regardless of the decomposition factor chosen.

It is worthwhile noting that the decompositions discovered are *refactored* for larger values of $k$. That is to say that the decomposition for $k = 5$ is not the same as the decomposition for $k = 4$ just with the addition of an extra subtask. Instead all five of the subtasks in the decomposition are adjusted allowing for maximum expressiveness in the representation. It follows that there is no intrinsic ordering of the subtasks. It only matters that they collectively form a good representation of the task space $Z$.

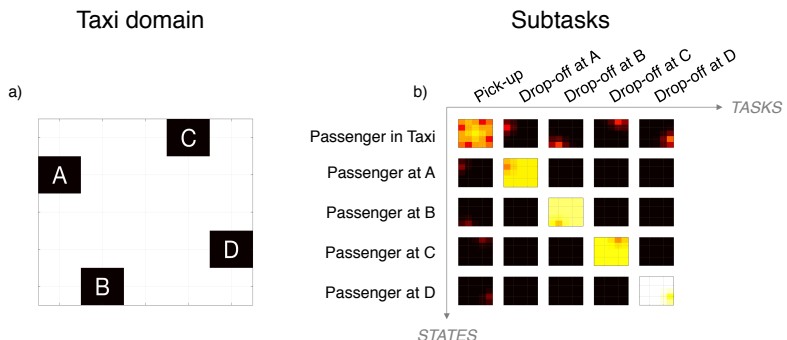

Figure 3: Subtasks discovered in the TAXI problem correspond to intuitive semantics despite the non-spatial nature of the domain. a) A variant of the standard TAXI domain in which a driver must navigate between four potential pick-up/drop-off locations in a $5 \times 5$ grid. b) Learned subtask decomposition. Each column corresponds to a subtask. The desirability functions for each subtask are overlaid onto a factored visualization of the domain in which each $5 \times 5$ block corresponds to the base domain with the passenger at a different location. The subtasks have been given intuitive names, although they are autonomously discovered.

While we have shown only spatial decomposition thus far, our scheme is applicable to tasks more general than simple navigation-like tasks. To make this point clear, we consider the scheme's application to the standard TAXI domain (Dietterich, 2000) with one passenger and four pick-up/drop-off locations. The $5 \times 5$ TAXI domain considered is depicted in Fig.(3-a). Here the agent operates in the product space of the base domain ($5 \times 5 = 25$), and the possible passenger locations (5-choose-1 $= 5$) for a complete state-space of 125 states. We consider a decomposition with factor $k = 5$. Fig.(3-b) is a depiction of the subtask structure we uncover.

Each column of Fig.(3-b) is one of the subtasks we discover. Each of these is a policy over the full state space. For visual clarity, these are then divided into the five copies of the base domain, each being defined by the passenger's location. The color-map corresponds to the desirability function for each subtask.

To help interpret the semantic nature of the subtasks discovered, consider the first column of Fig.(3-b). This subtask has almost all of its desirability function mass focused at states in which the passenger is in the Taxi. This task is thus a general pick-up action. By a similar analysis, column two of Fig.(3-b) depicts a subtask whose desirability function is essentially uniform over all states where the passenger is at location A. Semantically this subtask seeks to enter states with the passenger at location A regardless of taxi position. This subtask thus corresponds to the drop-off action at location A. Also note the slight probability leakage into the 'in taxi' state for the drop off point - the precondition for the passenger to be dropped off. Considered as a whole, the subtask basis represents policies for getting the passenger to each of the pick-up/drop-off locations, and for having the passenger in the taxi.

## 4 HIERARCHICAL DECOMPOSITIONS

The proposed scheme discovers a set of subtasks by finding a low-rank approximation to the desirability basis matrix $Z$. By leveraging the stacking mechanism defined in Saxe et al. (2017), this approximation procedure can simply be reapplied to find an approximate desirability basis for each subsequent layer of the hierarchy, by factoring the desirability matrix $Z^{l+1}$ at each layer. However,

as noted in section 2.1, in order to define the higher layer MLMDP in the first place, both the subtask states $S_t^{l+1}$, and the subtask passive dynamics $P_t^{l+1}$ must be specified.

The higher layer MLMDP will have $N_t^l = k^l$ states. Each of these states may be directly associated with the $k^l$ subtasks uncovered through the lower layer decomposition. Intuitively we have approximated the full task space with a $k^l$-dimensional subspace which captures maximal variance, and then formed a new MLMDP problem defined over this reduced space, as shown in Fig.(4).

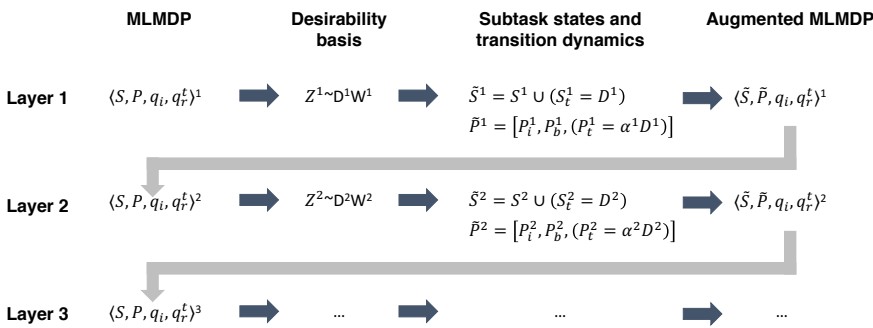

Figure 4: A simple recursive procedure for constructing hierarchical subtasks. At each layer, subtasks are uncovered by finding a low-rank approximation to the desirability basis $Z^l \approx D^l W^l$. Higher layers are formed autonomously by defining the subtask transition matrix as a scalar multiple of the data matrix, $P_t^l = \alpha^l D^l$. A designer need only specify the decomposition factors $k^l$.

As noted in section 2.1, the subtask passive dynamic $P_t^{l+1}$ is typically hand-crafted by a designer. While this is still possible as a design intervention, in a push for autonomous discovery, we relax this requirement and simply define the subtask transitions as

$$P_t^l = \alpha^l D^l, \tag{3}$$

where $\alpha^l$ is a single hand-crafted scaling parameter which controls how frequently the agent will transition to the higher layer(s). Defining $P_t^l$ in terms of $D^l$ has the effect of promoting transitions into the subtask states from nearby states in the lower layer, and demoting transitions into subtask states from far away states in the lower layer. This is intuitive as it ensures that our notional current state in the higher layer MLMDP closely represents our true base state.

As a demonstration of the recursive and multiscale nature of the scheme, we consider a spacial domain inspired by the multiscale nature of cities, see Fig.(5). At the highest level we consider a city which is comprised of three major communities, each of which is comprised of five houses. Each house is further comprised of four rooms, each of which is comprised of sixteen base states in a $4 \times 4$ grid. We consider a decomposition in line with the natural scales of the domain and take $k_l = \{3 \times 5 \times 4 = 60, 3 \times 5 = 15, 3\}$ respectively for $l = 2, 3, 4$. As expected, the scheme discovers subtasks corresponding to the multiscale nature of the domain with the highest layer subtasks intuitively corresponding to whole communities, etc. Of course the semantic clarity of the subtasks is due to the specific decomposition factors chosen, but any decomposition factors would work to solve tasks in the domain.

At this point the scheme has automated the discovery of the subtasks themselves, and the transitions into these subtasks. What remains is for a designer to specify the decomposition factors $k^l$ at each layer. In an analogy to neural network architectures, the scheme has defined the network connectivity but not the number of neurons at each layer. While this is a typical hyperparameter, by leveraging the unique construction in Eqn.(2), a good value for this parameter may be estimated from data. By increasing the decomposition factor $k^l$ the approximation error, given by Eqn.(2), is monotonically decreased. For some domains there is an obvious inflection point at which increasing the decomposition factor only slightly improves the approximation. Let us denote the dependence of $d_\beta(\cdot)$ on the decomposition factor simply as $f(k)$. Then we may somewhat naively take the smallest value that demonstrates diminishing incremental returns as a good value for $k$. In this instance the

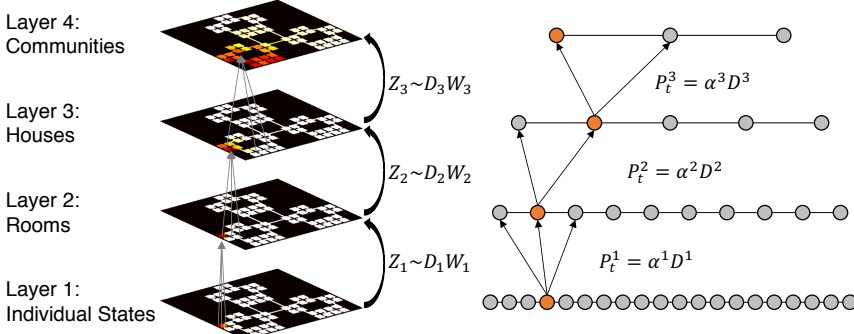

Figure 5: Recursive subtask discovery in a hierarchical domain. (LEFT) By projecting the state at each layer back into the base domain, it becomes apparent that subtasks correspond to distributed patterns of preferred states, rather than single goal states. In this way hierarchical subtasks are ever more abstracted in space and time, as higher layers are accessed. Tangibly, where the states at the lowest layer correspond to individual locations, higher layer states correspond to entire rooms, houses, and communities correspondingly. (RIGHT) An abstract representation of 'subtasks' as states of a higher layer MLMDP. A key contribution of this paper is to define an autonomous way of uncovering the contents of higher layer states, and the transition structures into these states.

approximation error, Eqn.(2), is said to exhibit elbow-joint behaviour:

$$\min_k \text{ s.t. } |f(k+1) - f(k)| < |f(k) - f(k-1)|. \tag{4}$$

In practice, when the task ensemble is drawn uniformly from the domain, the observed elbow-joint behaviour is an encoding of the high-level domain structure.

## 5 EQUIVALENCE OF SUBTASKS

Choosing the right set of subtasks is known to speed up learning and improve transfer between tasks. However, choosing the wrong subtasks can actually slow learning. While in general it is not possible to assess *a priori* whether a set of subtasks is 'good' or 'bad', the new approach taken here provides a natural measure of the quality of a set of subtasks, by evaluating the quality of the approximation in Eqn.(1). It follows immediately that different sets of subtasks can be compared simply by evaluating Eqn.(1) for each set individually. This leads naturally to the notion of subtask equivalence.

Suppose some standard metric is defined on the space of matrices as $m(A, B)$. Then a formal pseudo-equivalence relation may be defined on the set of subtasks, encoded as the columns of the data matrix $D$, by assigning subtasks that provide similar approximations to the desirability basis to the same classes. Explicitly, for $D_1, D_2, \in \mathbf{R}^{m \times k}$ we have $D_1 \sim D_2$ iff $m(Z - D_1 W_1, Z - D_2 W_2) < \epsilon$. The pseudo-equivalence class follows as:

$$[D_j] = \{D_i \in \mathbf{R}^{m \times k} \mid D_i \sim D_j\}. \tag{5}$$

A full equivalence relation here fails since transitivity does not hold.

### 5.1 ROOMS VERSUS DOORWAYS

As noted above, our scheme typically uncovers subtasks as complex distributions over preferred states, rather than individual states themselves. As in Fig.(2), we uncover regions such as 'rooms', whereas other methods typically uncover single states such as 'doorways'. There is a natural duality between these abstractions, which we consider below.

A weight vector can be assigned to each state by solving Eqn.(1) for a specific $z$:

$$w_s = \min_w ||z_s - Dw||^2. \tag{6}$$

This weight vector can be thought of as the representation of $s$ in $D$. To each state we then assign a real-valued measure of *stability*, by considering how much this representation changes under state-transition. Explicitly, we consider the stability function $g : S \to \mathbf{R}$:

$$g(s) = \sum_i p_{is} ||w_i - w_s||_2^2, \tag{7}$$

which is a measure of how the representations of neighbour states differ from the current state, weighted by the probability of transitioning to those neighbours. States for which $g(s)$ takes a high value are considered to be *unstable*, whereas states for which $g(s)$ takes a small value are considered to be *stable*. Unstable states are those which fall on the boundary between subtask 'regions'. A cursory analysis of Fig.(6) immediately identifies doorways as being those unstable states.

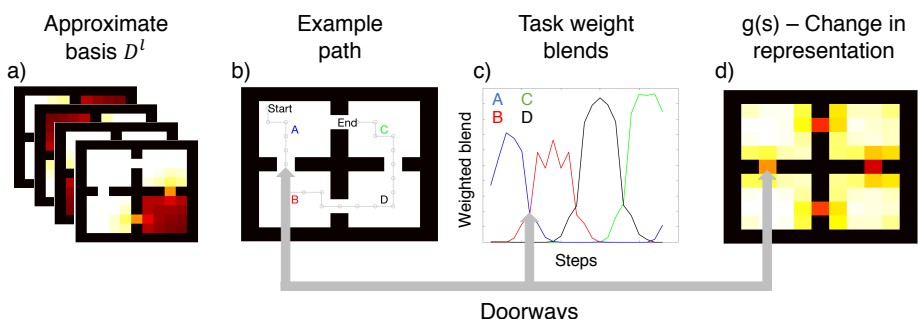

Figure 6: A natural duality exists between the subtasks uncovered by our scheme, and those typically uncovered by other methods. a) A filter-stack of four subtasks corresponding to the layer one decomposition. Here $k^1 = 4$ and we present the full set of subtasks. Each of the four subtasks corresponds to one of the four rooms in the domain. b) A hand picked example path through the domain, chosen to illustrate the changing representation for different domain states in terms of the higher layer states. This path and does not correspond to a real agent trajectory. c) For each state along the example path we compute the desirability function $z_s$ and approximate it using a linear blend of our subtasks according to Eqn.(6). The task weights are plotted as a function of steps, revealing the change in representation for different states along the example path. d) Agnostic to any particular path, we compute the stability function $g(s)$ for each state in the domain. It is immediately clear that *unstable* states, those for which the representation in $D^l$ changes starkly, correspond to 'doorways'.

## 6 Conclusion

We present a novel subtask discovery mechanism based on the low rank approximation of the desirability basis afforded by the LMDP framework. The new scheme reliably uncovers intuitive decompositions in a variety of sample domains. Unlike methods based on pure state abstraction, the proposed scheme is fundamentally dependent on the task ensemble, recovering different subtask representations for different task ensembles. Moreover, by leveraging the stacking procedure for hierarchical MLMDPs, the subtask discovery mechanism may be straightforwardly iterated to yield powerful hierarchical abstractions. Finally, the unusual construction allows us to analytically probe a number of natural questions inaccessible to other methods; we consider specifically a measure of the quality of a set of subtasks, and the equivalence of different sets of subtasks.

A current drawback of the approach is its reliance on a discrete, tabular, state space. Scaling to high dimensional problems will require applying state function approximation schemes, as well as online estimation of $Z$ directly from experience. These are avenues of current work. More abstractly, the method might be extended by allowing for some concept of nonlinear regularized composition allowing more complex behaviours to be expressed by the hierarchy.

### Acknowledgments

AMS thanks the Swartz Program in Theoretical Neuroscience at Harvard University for support.

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
