# OpenReview forum: "Hierarchical Subtask Discovery with Non-Negative Matrix Factorization"
_ICLR.cc/2018/Conference — Accept (Poster)_

### Official Review · AnonReviewer3 · 2017-11-30
**Novel model to discover subtasks in probabilistic planning (LMDP).**

**Rating:** 6
**Confidence:** 2

**Review:**

This paper proposes a formulation for discovering subtasks in Linearly-solvable MDPs. The idea is to decompose the optimal value function into a fixed set of sub value functions (each corresponding to a subtask) in a way that they best approximate (e.g. in a KL-divergence sense) the original value.

Automatically discovering hierarchies in planning/RL problems is an important problem that may provide important benefits especially in multi-task environments. In that sense, this paper makes a reasonable contribution to that goal for multitask LMDPs. The simulations also show that the discovered hierarchy can be interpreted. Although the contribution is a methodological one, from an empirical standpoint, it may be interesting to provide further evidence of the benefits of the proposed approach. Overall, it would also be useful to provide a short paragraph about similarities to the literature on discovering hierarchies in MDPs.

A few other comments and questions:

- This may be a fairly naive question but given your text I'm under the impression that the goal in LMDPs is to find z(s) for all states (and Z in the multitask formulation). Then, your formulation for discovery subtasks seems to assume that Z is given. Does that mean that the LMDPs must first be solved and only then can subtasks be discovered? (The first sentence in the introduction seems to imply that there's hope of faster learning by doing hierarchical decomposition).

- You motivate your approach (Section 3) using a max-variance criterion (as in PCA), yet your formulation actually uses the KL-divergence. Are these equivalent objectives in this case?


Other (minor) comments:

- In Section it would be good to define V(s) as well as 'i' in q_i (it's easy to mistake it for an index).

---

> ### Author Response · Authors · 2018-01-05
> **Response to Reviewer 3**
>
> We would like to thank the reviewer for their efforts and insightful comments.
>
> Similarities to other hierarchical discovery methods:
> Where most other approaches have been used to learn a single level of hierarchy. Our method is distinctive mainly in being able to be iterated repeatedly, forming deep hierarchies. We have extended our discussion of this point in the paper.
>
> The paper assumes that the multitask Z matrix is given:
> We assume that Z is given for a basis set of tasks, not for all possible tasks. There are a number reasons that we believe this is not a limiting assumption:
> 1.	The basis set of tasks can be a tiny fraction of the set of possible tasks in the space. As an example, suppose we consider tasks with boundary rewards at any of two separate locations in an N dimensional world such that there are N-choose-2 possible tasks (corresponding to tasks like “navigate to point A or B”). We require only an N-dimensional Z matrix containing tasks to navigate to each point individually. The resulting subtasks we uncover will aid in solving all of these N-choose-2 tasks. More generally we might consider tasks in which boundary rewards are placed at three or more locations, etc. To know Z therefore means to know an optimal policy to achieve N of ~2^N tasks in a space.
> 2.	While we assume knowledge of Z in the paper, we needn’t have a full N-task Z matrix for the method to applied as is. Suppose we had a smaller Z_hat matrix corresponding to M<N tasks. The method would nevertheless find a compressed representation for those M tasks and in so doing uncover useful subtasks. We assume the full Z matrix in the paper so that uncovered subtasks are intuitive decompositions of the full state space. If we consider Z_hat with tasks drawn only from some subsection of the state space, our method would uncover a compressed representation of just the subspace (in this way our method can be said to be task dependent). Similarly if we consider Z_hat with tasks drawn uniformly over the state space we uncover similar decompositions to those presented in the paper.
> 3.	Ultimately, in practice we would like to obtain estimates for Z online from experience in a domain. This could be done either directly through Z-iteration (an off-policy value iteration-like update), or by first building a state transition model. Methods to achieve this sort of estimate are well understood. We believe the results presented in this work are a necessary precursor: before tackling the joint estimation of Z and the hierarchy, we wanted to focus solely on inferring the hierarchy, which is in our view a critically challenging aspect of the problem. Just as LMDPs were first solved in batch mode as an eigenvalue problem before developing online z-iteration, we wanted to formulate and solve the computational problem in the batch setting before turning to online learning. Online learning thus is beyond the scope of this paper, though it is a focus of our current work, and we are excited to see this in the near future.
> We have revised the paper to make this point more strongly.
>
> Equivalence of maximum-variance and kl-divergence in the matrix factorization:
> We do not believe that the maximum-variance (\beta=2) and the kl-divergence (\beta=1) are mathematically equivalent. Instead the intuition for the decomposition scheme came from a maximum variance like argument, but the kl-divergence cost was ultimately chosen in practice to align with the RL objective cost for LMDPs. In practice the method does not appear to be overly sensitive to the choice of \beta in the range [1,2].  For extreme values of \beta outside this range results degrade.
>
> The empirical value of our method:
> The fact that a task hierarchy can yield efficiency improvements in the multitask setting was shown in (Saxe et al., 2017, ICML). In this instance the hierarchy was, however, hand crafted. More generally, when any ‘good’ hierarchy is provided (one in which new tasks case be represented well within the hierarchy), the learning jump-start is observed.
> A full investigation into the empirical value of the method in the online setting is very interesting, but it is beyond the scope of the present submission.

---

### Official Review · AnonReviewer4 · 2017-12-08
**Further development needed before the method can be applied in RL**

**Rating:** 5
**Confidence:** 2

**Review:**

The paper builds upon the work of Saxe et al on multitask LMDP and studies how to automatically discover useful subtasks. The key idea is to perform nonnegative matrix factorization on the desirability matrix Z to uncover the task basis.

The paper does a good job in illustrating step by step how the proposed algorithms work in simple problems. In my opinion, however, the paper falls short on two particular aspects that needs further development:

(1) As far as I can tell, in all the experiments the matrix Z is computed from the MDP specification. If we adopt the proposed algorithm in an actual RL setting, however, we will need to estimate Z from data since the MDP specification is not available. I would like to see a detailed discussion on how this matrix can be estimated and also see some RL experiment results.

(2) If I understand correctly, the row dimension of Z is equal to the size of the state space, so the algorithm can only be applied to tabular problem as-is. I think it is important to come up with variants of the algorithm that can scale to large state spaces.

In addition, I would encourage the authors to discuss connection to Machado et al. Despite the very different theoretical foundations, both papers deal with subtask discovery in HRL and appeal to matrix factorization techniques. I would also like to point out that this other paper is in a more complete form as it clears the issues (1) and (2) I raised above. I believe the current paper should also make further development in these two aspects  before it is published.

Minor problems:
- Pg 2, "... can simply be taken as the resulting Markov chain under a uniformly random policy". This statement seems problematic. The LMDP framework requires that the agent can choose any next-state distribution that has finite KL divergence from the passive dynamics, while in a standard MDP, the possible next-state distribution is always a convex combination of the transition distribution of different actions.

References
Machado et al. ICML 2017. A Laplacian Framework for Option Discovery in Reinforcement Learning.

---

> ### Author Response · Authors · 2018-01-05
> **Response to Reviewer 4**
>
> We would like to thank the reviewer for their efforts and insightful comments.
>
> How can we estimate Z from data:
> Estimating Z from data in an online RL setting is an important question and is the focus of our current research efforts. The simplest approach is to apply online z-iteration, an off-policy form of value iteration. Z-iteration is a state-based scheme, and because it is off-policy, all tasks in the Z matrix can be updated regardless of which task is currently being executed. An alternative approach is to obtain estimates for the transition model, and then solve for Z. However, we emphasize that the RL setting is not the only possible application of our method: even without online RL, our method allows the automatic discovery of hierarchy and its use in planning in a batch or offline setting. We note that other prior methods like the Bayesian estimation approach of Solway et al., 2014 (“Optimal behavioral hierarchy,” PLoS Comp Bio) or the information theoretic approach of McNamee et al., (“Efficient state-space modularization for planning,”
> NIPS 2016) do not operate in the online RL setting, require full knowledge of the state space and transition model, and operate only in tabular representations. These algorithms have still been fundamental in specifying the computational problem to be solved. Our method goes beyond this prior work most significantly by being able to learn multiple levels of hierarchy, and being more computationally efficient. We believe these features make this work significant for the offline setting. Given that the online RL setting introduces additional variables, we have elected to take a more gradual approach to the development of the method - first ensuring that the new subtask discovery concepts are robust in tabular batch settings before tackling the joint estimation of Z and the hierarchy in high dimensional problems. We are very keen to see an empirical demonstration of online learning in this new framework soon, but it is beyond the scope of the present submission.
>
> Variants of the algorithm to allow for problems with non-tabular state representations:
> This will certainly be an important extension of the method, and again, is the focus of current work. As it stands, current approaches to deep RL use function approximation over the state space, but keep a tabular representation of tasks. Conversely, our approach is, at present, a tabular representation over states but function approximation over tasks.
> When the number of tasks one wishes to perform in some space is significantly smaller than the state space, current approaches seem sensible. On the other hand, when the number of tasks we wish to perform is much greater than the number of states, current approaches appear unlikely to scale well. We believe filling in this possibility—demonstrating how function approximation can be safely used to perform many tasks organized hierarchically—is an important contribution. Ultimately, we must find a way to combine the best of both worlds and do function approximation both in the state and task space, but this is beyond the scope of the present submission.
>
> Discuss comparisons to Machado et al.:
> Thank you for the important pointer, we now include a discussion comparing our approach with that in Machado et al (2017). As noted, while both papers are concerned with options discovery, and utilize matrix factorization tools to achieve this, they have different theoretical foundations and yield different results in practice. Their approach to extend the core concepts therein to a linear function approximation scheme is instructive, and will be useful to our current work.
> There are several notable differences between these methods. Most importantly, our method can be recursively applied to generate arbitrarily deep control hierarchies, while it is not immediately clear (and there has been no empirical demonstration of) how the approach taken in Machado et al. might achieve a deeper hierarchical architecture, or enable immediate generalization to novel tasks.
> The methods appear to some extent to be orthogonal (with one supporting function approximation techniques in the state space, and the other supporting deep hierarchies and function approximation in the task space), and thus could potentially be profitably combined.

---

> > ### Author Response · Authors · 2018-01-05
> > **Response to Reviewer 4 (continued)**
> >
> > Taking the passive dynamics to be the uniform random policy:
> > A uniform random policy for the passive dynamics is a common choice in LMDPs which is suitable for a variety of tasks (spatial navigation, trajectory planning, Tower of Hanoi, etc). This is, however, a modeling assumption and there is no requirement that the passive dynamics be derived from the uniform random policy. Choosing some alternate reference policy is possible, and may be suitable for specific tasks. More generally, Todorov (2009) provides a general way of approximating MDPs using LMDPs. We have added citations to a variety of works which show how standard domains have been modeled in the LMDP framework.

---

### Official Review · AnonReviewer5 · 2017-12-12
**Extends previous work that considered multitask learning in RL and proposed a hierarchical learner based on concurrent execution of many actions in parallel. Current work proposes an autonomous way to build the hierarchy and discover subtasks.**

**Rating:** 7
**Confidence:** 3

**Review:**

The present paper extends a previous work by Saxe et al (2017) that considered multitask learning in RL and proposed a hierarchical learner based on concurrent execution of many actions in parallel. That framework made heavy use of the framework of linearly solvable Markov decision process (LMDP) proposed by Todorov, which allows for closed form solutions of the control due to the linearity of the Bellman optimality equations. The simple form of the solutions allow them to be composed naturally, and to form deep hierarchies through iteration. The framework is restricted to domains where the transitions are fixed but the rewards may change between tasks.  A key role is played in the formalism by the so-called ‘free dynamics’ that serves to regularize the action selected.

The present paper goes beyond Saxe et al. in several ways. First, it renders the process of deep hierarchy formation automatic, by letting the algorithm determine the new passive dynamics at each stage, as well as the subtasks themselves. The process of subtask discovery is done via non-negative matrix factorization, whereby the matrix of desirability functions, determined by the solution of the LMDPs with exponentiated reward. Since the matrix is non-negative, the authors propose a non-negative factorization into a product of non-negative low rank matrices that capture its structure at a more abstract level of detail. A family of optimization criteria for this process are suggested, based on a subclass if Bregman divergences. Interestingly, the subtasks discovered correspond to distributions over states, rather than single states as in many previous approaches. The authors present several demonstrations of the intuitive decomposition achieved. A nice feature of the present framework is that a fully autonomous scheme (given some assumed parameter values) is demonstrated for constructing the full hierarchical decomposition.

I found this to be an interesting approach to hierarchical multitask learning, augmenting a previous approach with several steps leading to increased autonomy, an essential agent for any learning agent. Both the intuition behind the construction and the application to test problem reveal novel insight. The utilization on the analytic framework of LMDP facilitates understanding and efficient algorithms.

I would appreciate the authors’ clarification of several issues. First, the LMDP does not seem to be completely general, so I would appreciate a description of the limitations of this framework. The description of the elbow-joint behavior around eq. (4) was not clear to me, please expand. The authors do not state any direct or indirect extensions – please do so. Please specify how many free parameters the algorithm requires, and what is a reasonable way to select them. Finally, it would be instructive to understand where the algorithm may fail.

---

> ### Author Response · Authors · 2018-01-05
> **Response to Reviewer 5**
>
> We would like to thank the reviewer for their efforts and insightful comments.
>
> Limitations of the LMDP framework:
> The LMDP framework on the surface appears very different from the standard MDP setting, and the question of its limitations arises frequently. In our view the LMDP framework is in fact quite general, and can be used to solve non-navigational and conceptual tasks such as the TAXI domain, and the Towers of Hanoi problem. For more on the generality of the LMDP see (Saxe et al., ICML 2017 supplementary material), which describes ways in which a variety of tasks have been modeled as LMDPs. The initial work of Todorov, 2009, for instance, gives a method for approximating any MDP with an LMDP. The main limitation of the LMDP framework, so far as we understand it, is that actions must incur costs: the transition cost in LMDPs necessarily has a KL divergence term with respect to the passive dynamics, which is non-negative. Hence, for instance, it must be costly to move from one position in a grid to the next (more precisely, to deviate from the passive dynamics). The LMDP would struggle to model a situation in which actions have strong rewards, eg, where the goal is to take the most circuitous path to a destination. We do not view this as a strong limitation, however, since nearly all domains have a principle of efficient action and it is common to place costs on each action taken in a traditional MDP. Indeed, we would argue that the LMDP exploits this shared structure in nearly all real-world tasks to allow more efficient solutions.
>
> Parsing the phrase “elbow-joint behavior”:
> One of the hyper parameters in our method is the number of nodes/subtasks at each level of the hierarchy. This corresponds to the rank of the decomposition. This choice is akin to choosing the number of neurons at different layers of a NN. Nevertheless we make an observation that may provide a way to establish a good value for even this parameter choice from data.
> The key idea is that by increasing the rank of the decomposition we monotonically improve the approximation to Z, as the error ||Z-DW|| tends to zero. For some domains there is an obvious inflection point at which increasing the rank of the decomposition only slightly improves the approximation. This suggests a natural trade-off between expressiveness of the hierarchy, and the additional computational effort required to support additional subtasks. When we plot the quality of the approximation, ||Z-DW||, against the decompositions factor k, the observed inflection point is described as exhibiting “elbow-joint” behavior. We have clarified this point in the text.
>
> Extensions and future work:
> We view this work as being an important stepping stone on the path towards a method for fully online learning of a deep control hierarchy. In that vein, there are a number of natural extensions (a few of which were rightly called out by other reviewers). Some of the major items are:
> 1.	Estimating Z from data (either directly or by learning a transition model), so that the agent can operate completely online
> 2.	Introducing standard notions of function approximation and compressed state representations to allow the method to scale to high dimensional state spaces
> 3.	Introducing some concept of regularized nonlinear composition; allowing more complex behavior to be approximated by the hierarchyMany of these items are the focus of our current research efforts.
>
> Free parameters and how to specify them:
> The number of hyper parameters introduced by our method is minimal.
> 1.	The number of nodes/subtasks at each level of the hierarchy
> ◦	This is a common set of hyper parameters for many deep learning applications
> ◦	The elbow joint behavior provides one possible path to estimate efficient values here from data
> ◦	In practice we choose the number of nodes at layer (l+1) to be approximately log(|S^l|), where |S^l| is the number of states at the preceding layer. This also determines the number of layers
> 2.	The subtask transition matrix Pt contains a scaling parameter such that Pt = \alpha W. Here \alpha controls how frequently the agent will consult the hierarchy for guidance. In practice we chose \alpha ~0.2. The intuition here is that it is important that the agent to be able to consult the hierarchy sufficiently frequently that it influences its behavior; but overly frequent access wastes computational resources.
> 3.	The choice of \beta in the cost function for the matrix decomposition. In our experiments we have typically chosen \beta = 1 (KL) or \beta = 2 (Maximum Variance). All of our experiments suggest that the method is not overly sensitive to choices for \beta in the range [1,2]. For extreme values of \beta outside this range results degrade.

---

> > ### Author Response · Authors · 2018-01-05
> > **Response to Reviewer 5 (continued)**
> >
> > Where does the algorithm fail:
> > This is a great question. We believe the method fails most obviously in domains in which there is no latent structure to abstract. For example, if the passive dynamics (at any level) are fully connected and uniform, then the decomposition delivers no value. While such a problem is degenerate in the base case, it is not yet clear to us under what conditions the recursive iteration of the hierarchical abstraction might at some point yield such a uniform structure (rendering further recursion useless).

---

### Decision · Program_Chairs · 2018-01-29
**ICLR 2018 Conference Acceptance Decision**

**Decision:**

Accept (Poster)

**Comment:**

Overall this paper seems to make an interesting contribution to the problem of subtask discovery, but unfortunately this only works in a tabular setting, which is quite limiting.